# Overweight or Obesity among Chinese Han People with Schizophrenia: Demographic, Clinical and Cognitive Correlations

**DOI:** 10.3390/brainsci13091245

**Published:** 2023-08-26

**Authors:** Xiao Huang, Yuan Sun, Anshi Wu, Xiangyang Zhang

**Affiliations:** 1Department of Anesthesiology, Beijing Chao-Yang Hospital, Capital Medical University, Beijing 100020, China; huanghuang94@yeah.net (X.H.);; 2Department of Pharmacy, Peking University Third Hospital, Beijing 100191, China; sunny5106@163.com; 3CAS Key Laboratory of Mental Health, Institute of Psychology, Beijing 100101, China; 4Department of Psychology, University of Chinese Academy of Sciences, Beijing 101408, China

**Keywords:** schizophrenia, Chinese Han, demographics, overweight, obesity, cognition

## Abstract

People with schizophrenia are more likely to be afflicted by obesity or overweight compared to the general population. This study aimed to explore the incidence of overweight and obesity, clinical features and cognitive performance of Chinese Han patients with chronic schizophrenia who had overweight or obesity. We obtained data from 985 schizophrenia inpatients about overweight and obesity through body mass index (BMI). All patients were evaluated with the positive and negative syndrome scale, the Mini-mental State Examination (MMSE) and the repeated battery for evaluation of the neuropsychological status (RBANS) scale. We collected demographic and clinical data using self-reported questionnaires. We divided patients into normal weight (BMI < 24 kg/m^2^), overweight (24 ≤ BMI < 28 kg/m^2^) and obese (≥28 kg/m^2^) groups according to the Working Group on Obesity in China (WGOC) criteria. We compared the clinical data between the three groups and then conducted binary logistic regression and linear regression to assess variables that were significantly associated with overweight and obesity and higher BMI. Of the sample, 324 (32.9%) and 191 (19.4%) patients had overweight and obesity, respectively. Patients who had overweight and obesity were younger, had less education, had higher waist and hip circumferences, higher rates of diabetes and a higher sumPANSP score (compared with patients in the normal group, *p* < 0.05). There were more female patients with obesity (compared with patients in the normal and overweight groups, *p* < 0.05). Logistic regression analysis indicated that overweight and obesity were associated with sumPANSP (OR = 1.03, 95%CI = 1–1.061, *p* = 0.049) and diabetes (OR = 1.891, 95%CI = 1.255–2.849, *p* = 0.002). Further linear regression showed that age (B = −0.004, *t* = −2.83, *p* = 0.005), educational level (B = −0.037, *t* = −2.261, *p* = 0.024), diabetes (B = 0.133, *t* = 2.721, *p* = 0.007) and sumPANSP (B = 0.008, *t* = 2.552, *p* = 0.011) were risk factors for higher BMI. We did not find cognitive performance differences between patients with or without overweight and obesity. Overweight and obesity were associated with some demographic and clinical factors in patients with persistent schizophrenia.

## 1. Introduction

Schizophrenia is a serious mental illness with a global lifelong prevalence of approximately 0.4% [1]. It has a damaging effect on patients and their caregivers and entails significant costs to the healthcare system [2]. Schizophrenia usually presents with positive symptoms, negative symptoms and cognitive impairment [3,4]. The mortality rate of people with schizophrenia is several times higher than that of the general population [5,6].

Being overweight and obese are risk factors for many chronic diseases, including cardiovascular disease, hypertension and microvascular disease [7]. People with schizophrenia are more likely to be afflicted by obesity or overweight compared to the general population [8]. However, the prevalence of overweight and obesity in people with schizophrenia varies worldwide. For example, Kassem et al. found that in Lebanon, 59.1% of male people with schizophrenia were obese/overweight. Older age and higher negative and total positive and negative syndrome scores were significantly correlated with lower body mass index (BMI) levels [9]. Chouinard et al. demonstrated that the prevalence of overweight and obesity in patients and outpatients with schizophrenia, schizoaffective disorder, and bipolar disorder in the United States were 29.4% and 33.2%, respectively, with an overall prevalence of 62.6% [10]. The high prevalence of overweight or obesity in people with schizophrenia is related to the adverse effects of antipsychotics, pre-treatment/pre-morbid genetic vulnerability, psychosocial and socioeconomic risk factors, and unhealthy lifestyles [10]. Lifestyle and environmental factors, psychotropic drugs, genetics and neuroendocrine processes play an important role in the association between schizophrenia and overweight or obesity [11].

Cross-cultural research is crucial to better understanding overweight and obesity [12]. For example, diverse environmental factors, genetic factors and economic reasons are the main causes of overweight and obesity [13]. Studies focused on overweight and obesity in people with schizophrenia are still limited. There is accumulating evidence that obesity is correlated with cognitive dysfunction in schizophrenic populations [14,15]. People with schizophrenia present with moderate to severe cognitive deficits, especially in the areas of attention, executive function, memory, language skills and processing speed deficits [16]. Cognitive deficits may be a central pathophysiological characteristic of the disorder. However, whether people with schizophrenia with overweight or obesity have serious cognitive decline is still unclear. Additionally, we have no idea about the prevalence and risk factors of overweight and obesity in Chinese Han people with schizophrenia. Therefore, we performed the repeated battery for evaluation of the neuropsychological status (RBANS) to confirm the hypothesis that worse cognitive performance would be found in people with schizophrenia with overweight or obesity in this study. We included a large sample of people with schizophrenia to further explore the prevalence, features, and cognitive performance of overweight or obesity in a Chinese Han population. The study presented here will explore (1) the prevalence of overweight and obesity in Chinese Han people with schizophrenia and (2) cognitive function and the clinical correlates in these specific patients. We proposed the following hypotheses: (1) the high prevalence of overweight and obesity in Chinese Han people with schizophrenia; (2) people with schizophrenia with overweight and obesity exhibit greater cognitive impairment cognition than people with schizophrenia with normal BMI; and (3) there would be some clinical correlates and risk factors in people with schizophrenia with overweight and obesity.

## 2. Methods

### 2.1. Subjects

We included chronic inpatients admitted From January to June 2019. The review board of the Institute of Psychology, Chinese Academy of Sciences (H18031) approved the study. Every included patient signed a written informed consent. The inclusion criteria were (1) age between 18–70 years; (2) diagnosis of schizophrenia by the DSM-IV structured clinical interview by two independent experienced psychiatrists [17]; (3) a minimum illness duration of 1 year; (4) had received a regular dose of oral antipsychotics more than 6 months before enrollment; (5) and would be able to provide informed consent prior to enrollment. The patients’ medical data were obtained and physical examinations were conducted. Patients with medical commodities and alcohol or substance abuse/dependence were excluded from the study.

Antipsychotics administered to patients consisted of clozapine (n = 239); risperidone (n = 194); olanzapine (n = 78); aripiprazole (n = 56); quetiapine (n = 31); sulpiride (n = 20); amisulpride (n = 14); ziprasidone (n = 7); perphenazine (n = 6); chlorpromazine (n = 4); and other (n = 8). The usage of antipsychotics in eight patients was not documented in this study. Daily doses of antipsychotics were converted to equivalent doses of chlorpromazine. The daily dosage of antipsychotics in all patients was 249.2 ± 124.6 mg/day.

### 2.2. BMI

In the present study, BMI was calculated as weight (kg) divided by height (m) square (calibrated to 0.1 kg using an electronic scale eb9003l, Xiangshan, China). All patients were divided into normal weight (BMI < 24 kg/m^2^), overweight (24 ≤ BMI < 28 kg/m^2^) and obese (≥28 kg/m^2^) according to the Working Group on Obesity in China (WGOC) criteria [18].

### 2.3. Sociodemographic Characteristics

The researchers conducted a detailed questionnaire to acquire information on sociodemographic characteristics, medical and psychological conditions, smoking behavior and medical history. We obtained data from the patients’ medical notes and clinicians.

The positive and negative syndrome scale (PANSS) was performed to measure symptoms, syndromes and severity of schizophrenia [19], which consists of positive symptoms, negative symptoms and general psychopathology subscales. The Insomnia Severity Index (ISI) measures the severity of insomnia through the previous month according to the self-report/recall [20]. It includes 7 individual items scored on a 5-point scale (e.g., each item is rated from 0–4) to reach a total score range from 0 to 28. A score of 0–7 represents no clinical insomnia, 8–14 represents subthreshold insomnia, 15–21 represents moderate insomnia and 22–28 represents severe insomnia.

Mini-mental state examination (MMSE) was used for cognition assessment. MMSE includes 5 subsections (orientation, registration and recall, attention and calculation, language and praxis). MMSE scores vary between 0–30, with lower scores suggesting poorer cognitive ability. Every question carries three possible answers: correct, incorrect and unanswerable. We calculate the unanswerable as the incorrect answer. MMSE uses a cutoff of 25 as a diagnosis of mild cognitive impairment (MCI) [21].

Two psychologists conducted the Repeatable Battery for the Assessment of Neuropsychological Status (RBANS, Form A) to assess cognition. Except for effective screening for dementia, it can also be conducted for a variety of disorders, including schizophrenia [22]. RBANS includes 12 subtests which resulted in five age-adjusted index scores and a total score. The five sections consist of attention, language, visuospatial/constructional, immediate memory, and delayed memory. RBANS is utilized for assessment when the patient is stable and does not present with psychiatric symptoms [23]. Zhong et al. showed its good clinical utility and dependability in schizophrenic patients [24]. RBANS showed excellent clinical efficacy and test–retest reliability in both people with schizophrenia and the general population.

The scales were evaluated by trained psychiatrists. The inter-rater correlation coefficient of the scales between the evaluators were all greater than 0.8. They assessed patients’ cognitive performance with RBANS on the day of or the following day of blood draws.

### 2.4. Statistical Analysis

This was a cross-sectional study and we assumed that the prevalence of overweight or obesity in patients with schizophrenia was 25% based on previous data from the literature [25,26]. We obtained a sample size of 832 with a two-sided α of 0.05 and a tolerance error of 3%. A minimum of 924 patients with schizophrenia was required when taking into account a 10% loss to follow-up rate. Finally, we included 985 people with schizophrenia.

DAG (directed acyclic graph) is useful to show the relation between variables in the cross-sectional study. The DAG makes it possible to draw theoretical diagrams of the links between variables and to identify which variables should be controlled in a multivariate model, thus preventing bias. We used DAG (Figure 1) to explore the association between schizophrenia and overweight and obesity and potential confounding factors. Using this graph, the software Dagitty indicated that the possible cofounders that should be adjusted were age and sex.

We performed the Shapiro–Wilk test and Q-Q plots to test the normality, and conducted the Levene test to confirm the equality of variances. Patients were divided into the normal group, overweight group and obesity group according to BMI. We expressed the incidence of overweight and obesity among people with schizophrenia as a percentage by the chi-square (χ^2^) test to compare the incidence between male and female patients. Analysis of variance and χ^2^ were performed to compare the variations in demographic and clinical characteristics between the three groups. Missing values are interpolated using mean values. The Bonferroni correction was performed to adjust for multiple tests. Multivariate analysis of covariance was conducted to investigate differences in cognitive status on the five and the total index scores of the RBANS, while adjusting for the potential confounding parameters (age, sex and educational years). Binary logistic regression was performed to assess variables that were significantly associated with overweight and obesity. Odds ratios (OR) resulted from logistic regression analyses to compare overweight and obesity among people with schizophrenia after correcting for associated variables. To better clarify the link between BMI levels and statistically significant indicators, we conducted multiple linear regression to assess variables that were significantly correlated with higher BMI. All analysis was performed in SPSS version 25.0. The statistical significance was set with *p* < 0.05.

## 3. Results

A total of 985 patients were included in the final analysis. Of the sample, 324 (32.9%) and 191 (19.4%) patients had overweight and obesity, respectively. The overweight and obesity rate for males and females was 35.7% (227/636) and 29.9% (97/349), respectively. The obesity rates for males and females were 14.8% (94/636) and 27.8% (97/349), respectively (χ^2^ = 24.416, *p* < 0.001, OR = 2.219). Table 1 expresses the sociodemographic and clinical characteristics of all the patients. The average age of the included patients was 47.2 ± 12.5 years old. The mean educational years and BMI were 9.2 ± 3.2 and 24.8 ± 4.6. The mean values of the total RBANS and MMSE scores for all patients were 78.6 ± 17.4 and 23.2 ± 6.2, respectively. Compared with patients without overweight or obesity, people with schizophrenia with overweight and obesity were younger, had lower education and had higher waist and hip circumference (both *p* < 0.01; Bonferroni corrected both *p* < 0.05). Additionally, people with schizophrenia with overweight and obesity were more likely to be female (χ^2^ = 25.191, *p* < 0.001) and have a higher rate of diabetes (χ^2^ = 6.311, *p* = 0.043) (only sex survived after Bonferroni corrected). Moreover, people with schizophrenia with overweight or obesity had higher sumPANSP than those without overweight and obesity (obesity: 17.74 ± 5.49 vs. overweight: 16.24 ± 5.8 vs. normal: 15.71 ± 4.87; d = 0.203, 95%CI = 0.078–0.329; *p* <0.001; Bonferroni corrected both *p* < 0.05). The daily dosage of antipsychotics in the normal, overweight and obese groups were 250 (145,410), 290 (150,404.4) and 240 (125,400) mg/day, respectively (Z = 2.859, *p* = 0.239).

RBANS data were available for 243 patients with overweight, 141 patients with obesity and 339 patients without overweight and obesity. The results of non-parametric and ANOVA tests applied between the three groups found that there were no significant differences between the five RBANS subtests and total scores (all *p* > 0.05). There remained no differences in RBANS cognitive performance scores between the three groups after adjusting for age, sex and educational years (all *p* > 0.05). We also did not find any significant difference in MMSE between the two groups (Z = 0.431, *p* = 0.806). (Table 2)

Stepwise logistics regression analysis illustrated that the following parameters were still significantly associated with overweight and obesity: sumPANSP (OR = 1.03, 95%CI = 1–1.061, *p* = 0.049) and diabetes (OR = 1.891, 95%CI = 1.255–2.849, *p* = 0.002) (Table 3). Further linear regression showed that age (B = −0.004, *t* = −2.83, *p* = 0.005), educational level (B = −0.037, *t* = −2.261, *p* = 0.024), diabetes (B = 0.133, *t* = 2.721, *p* = 0.007) and sumPANSP (B = 0.008, *t =* 2.552, *p* = 0.011) were risk factors for higher BMI (Table 4).

In this study, the main findings were as follows: (1) 324 (32.9%) and 191 (19.4%) patients had overweight and obesity, respectively. Compared with patients without overweight or obesity, people with schizophrenia with overweight and obesity were younger, had lower education and larger waist and hips, more likely to be female and had a higher rate of diabetes. Moreover, people with schizophrenia with overweight or obesity had higher sumPANSP than those without overweight and obesity. (2) No significant differences were found between the five RBANS subtests, total scores and MMSE scores. (3) SumPANSP and diabetes were associated with overweight and obesity in people with schizophrenia. Age, educational level, sumPANSP and diabetes were risk factors for higher BMI.

## 4. Discussion

### 4.1. The Prevalence of Overweight and Obesity

The present study reported sociodemographic and clinical variables of Chinese schizophrenia inpatients with overweight and obesity. Our study adds to the existing literature on the prevalence and factors related to overweight and obesity in schizophrenia inpatients in a Chinese Han population, finding that 32.9% and 19.4% percent of people with schizophrenia are overweight and obese, respectively. The prevalence of overweight and obesity has been reported in many studies, but the results varied. For example, a follow-up survey from the China health and retirement longitudinal study (Charles) (2011 and 2013–2015) showed 28.07% and 9.26% percent of the overweight and obese population for men and 35.03% and 16.84% for women, respectively [26]. A meta-analysis of Chinese children and adolescents from 1991 to 2015 by Guo et al. identified an increase in the prevalence of overweight and obesity from 5.0% and 1.7% in 1991–1995 to 11.7% and 6.8% in 2011–2015, with the highest prevalence of overweight in 2006–2010 and a higher prevalence in urban areas than in rural areas [27]. In Hunan, China, the overall prevalence of overweight significantly increased from 20.81% to 26.97% and obesity significantly increased from 4.09% to 7.13%, respectively, from 2013 to 2018 [25].

In this study, the prevalence of overweight and obesity in people with schizophrenia is higher than that of the general population. There were interacting relationships between obesity and overweight and schizophrenia. The reasons that our results are different from other studies may be sample characteristics and different measures used. For example, our study population focused on people with schizophrenia with a mean age of 47 years, implying an included population that is predominantly middle-aged. At the same time, age is an important factor in overweight and obesity [28]. Being obese or overweight may increase disease burden, increase stigma, decrease self-esteem and social functioning, and decrease self-management behaviors like adhering to medication regimens [29]. The discrepancies in overweight and obesity rates might also be due to other reasons such as genetics, culture, environment and medication treatment [30,31].

### 4.2. Gender Differences in Overweight and Obesity in People with Schizophrenia

Our study detected a higher proportion of women than men by analyzing the patients with overweight and obesity. In agreement with our findings, Gurpegui et al. also found that overweight and obesity were associated with the female gender in people with schizophrenia [32]. Li et al. showed that gender, type 2 diabetes and education level are risk factors for obesity in people with schizophrenia [33]. We speculate the reason women in our study were more likely to be associated with overweight and obese may be that the average age of our patients was 47 years. So, most women included in this study were climacteric, with the side effects of reproductive hormones and menopausal symptoms on obesity [34]. A deep exploration of Chinese people with schizophrenia in gender differences with the clinical factors of overweight and obesity will be needed. However, the mechanisms underlying the association between gender and education and overweight or obesity are not currently elucidated.

### 4.3. PANSS Score in People with Schizophrenia with Overweight and Obesity

Schizophrenia is commonly comorbid with depressive symptoms. Also, depression is related to overweight or obesity [35]. In the present study, PANSS scores for positive symptoms factor differed remarkably between those who had and did not have overweight or obesity, indicating that overweight and obesity may accelerate the psychopathological symptoms in the early stages of schizophrenia. Although this correlation was not sustained after the Bonferroni correction, we can still derive a trend from the difference. The results agreed with the finding from Yang et al., which showed significantly higher PANSS positive symptom subscale scores in people with schizophrenia with overweight and obesity [36]. But, our results were not consistent with Tian et al., who showed that overweight and obesity were negatively associated with the severity of psychiatric symptoms [37]. The conflicting outcome may be related to the different stages of the disease, the use of antipsychotic drugs, etc.

### 4.4. Cognitive Performance in People with Schizophrenia with Overweight and Obesity

Cognitive impairment is one of the main symptoms of schizophrenia and is associated with worse outcomes. The burden of anticholinergic and benzodiazepine should be maintained to a minimum taking into account the negative influence on cognitive functioning. In psychosocial interventions, cognitive restoration and physical exercise are suggested to deal with cognitive deficits in schizophrenia. Non-invasive brain stimulation techniques can be considered as an additional therapy [38]. Both cognitive training (CT) and aerobic exercise show potential for ameliorating cognitive deficits in schizophrenia. The combination of cognitive remediation and physical exercise is recommended as it could provide valuable improvements in cognitive and metabolic outcomes such as overweight and obesity. Also, aerobic exercise performed close to the time of cognitive repair may bring the brain to a state of neuroplasticity readiness, thus promoting cognitive function [39]. Combining physical exercise with cognitive remediation offered privileged advantages and quicker improvement when compared to cognitive remediation alone [40]. In the present study, no cognitive performance differences were found between the three subgroups, which were contrary to our assumptions. Previous studies have shown a strong correlation between overweight or obesity and cognition [41,42]. The difference may result in the difference in the diagnosis of cognition and the inclusion of the population. Therefore, subsequent surveys must be fully addressed.

### 4.5. Limitations

The strength of the study includes the large size of the included sample and the use of validated measures to assess the clinical and cognitive characteristics of participants. There is no doubt that our study has some deficiencies. Firstly, our study is a cross-sectional study, so we cannot speculate on the causal relationship between overweight or obesity and schizophrenia clinical characteristics. Secondly, our study did not include some important variables such as parents’ BMI, diet, income, etc. Hence, we could not assess the confounding effect of these important factors on combined overweight or obesity in people with schizophrenia. Further in-depth studies are necessary. Thirdly, because our participants were long-term inpatients with a longer duration of illness and more intense psychopathology, they were separate from typical psychiatric outpatients or psychiatric hospital patients. Therefore, our results cannot be generalized to outpatients or community patients. Fourthly, the proportion of men and women in our included patients was not equal (636 male and 349 female), so our study needs to be carefully generalized to the general public. Fifthly, because RBANS was designed to identify and characterize aberrant cognitive decline in older adults, it may be not the best choice for assessing cognitive performance in this study (since our included patients were predominantly middle-aged). Also, because of features of linguistic and mathematics assessment, the MMSE is biased against those with low education. Finally, patients’ obesity or overweight may be inaccurate by only one measurement of height and weight.

## 5. Conclusions

Overall, we showed that the prevalence of overweight and obesity in people with schizophrenia was as high as 32.9% and 19.4% percent, respectively. In this study, people with schizophrenia who were overweight or obese were younger, had lower education, were more likely to be female and had higher sumPANSP scores. However, no significant difference in cognition was found between those with and without overweight or obesity. Due to some methodological limitations, our findings need to be interpreted with caution. Further multicenter large-scale studies are necessary to provide more insight into the prevalence and clinical characteristics of combined overweight or obesity in people with schizophrenia.

## Figures and Tables

**Figure 1 brainsci-13-01245-f001:**
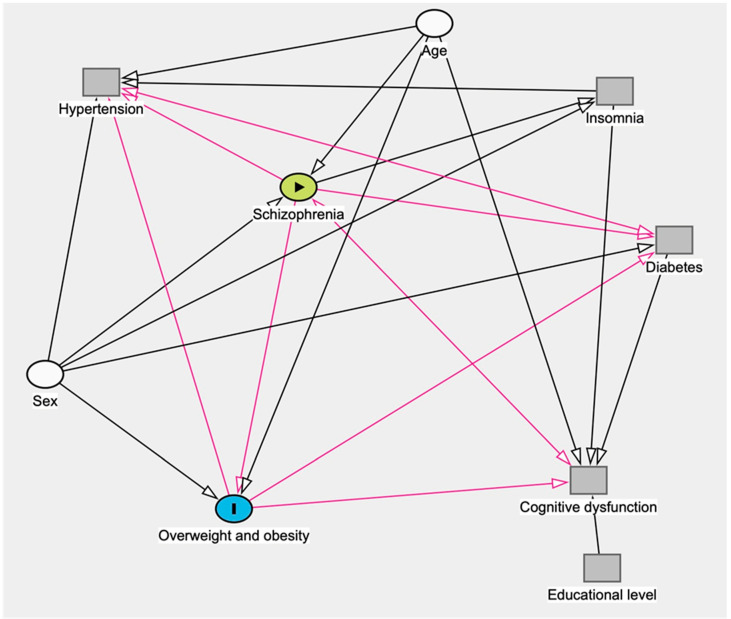
Directed Acyclic Graph (DAG) of the relationship schizophrenia and abesity and overweight. Adjusted variables (with circle). Based on DASitty version 3.0. black arrows: biasing path; pink arrows: causal path.

**Table 1 brainsci-13-01245-t001:** Demographic characteristics of people with schizophrenia who are overweight, obese or normal, N (%), M (SD) or Median (quartile).

	Normal (470)	Overweight (324)	Obesity (191)	F, Z or χ^2^	*p*
Age, year	47.96 (13.19)	47.13 (11.59)	45.27 (12.33)	3.143	0.044
Education level	2 (2,3)	2 (2,3)	2 (2,3)	7.022	0.03
Weight, kg	58.34 (7.64)	70.91 (6.87)	83.84 (12.09)	656.858	<0.001
Height, cm	165.73 (7.65)	165.69 (7.69)	162.46 (9.97)	12.194	<0.001
Waist, cm	84 (77,90)	94 (89,99)	100 (97,110)	362.197	<0.001
Hip, cm	92 (88,97)	99 (96,102)	106 (102.5,110)	341.617	<0.001
Gender, n (%)				25.191	<0.001
Male	315 (67)	227 (70.1)	94 (49.2)		
Female	155 (33)	97 (29.9)	97 (50.8)		
Marriage, n (%)				10.558	0.228
Single	294 (62.6)	184 (56.8)	103 (53.9)		
Married	95 (20.2)	80 (24.7)	53 (27.7)		
Divorced	67 (14.3)	56 (17.3)	32 (16.8)		
Widowed	11 (2.3)	3 (0.9)	3 (1.6)		
Habitat, n (%)				0.822	0.663
Ward	468 (99.6)	321 (99.1)	190 (99.5)		
Family	2 (0.4)	3 (0.9)	1 (0.5)		
Diabetes, n (%)	52 (11.1)	56 (17.3)	27 (14.1)	6.311	0.043
Hypertension, n (%)	61 (13)	51 (15.7)	36 (18.8)	3.858	0.145
Insomnia, n (%)				5.895	0.207
None	424 (90.2)	287 (88.6)	173 (90.6)		
Subthreshold insomnia	40 (8.5)	25 (7.7)	15 (7.9)		
Moderate to severe insomnia	6 (1.3)	12 (3.7)	3 (1.6)		
PANSS score					
Positive symptoms	15.71 (4.87)	16.24 (5.8)	17.74 (5.49)	9.548	<0.001
Negative symptoms	21 (17,25)	20 (16,23)	19 (15,25)	4.484	0.106
General psychopathology	40.43 (8.47)	40 (8.78)	41.3 (9.17)	1.293	0.275
Total	78.49 (16.78)	77.71 (17.77)	80.22 (18.41)	1.205	0.3
Systolic blood pressure	120 (115,126)	120 (119,126)	123 (118,130)	9.298	0.01
Diastolic blood pressure	78 (71.5,80)	80 (73,80)	78 (74,82)	1.748	0.417

Note: PANSS: positive and negative syndrome scale; educational level: 1—elementary school; 2—junior high school; 3—secondary high school; 4—undergraduate; 5—master’s degree; 6—doctorate; 7—uneducated.

**Table 2 brainsci-13-01245-t002:** Comparisons of MMSE and RBANS of neuropsychological status scores between patients with chronic schizophrenia who are overweight, obese or normal.

	Normal	Overweight	Obesity	F, Z or χ^2^	*p*
MMSE score	26 (22,28)	26 (24,28)	26 (23,28)	0.431	0.806
Cognitive impairment, n (%)				3.499	0.744
None	163 (36.2)	125 (40)	67 (36.8)		
Mild cognitive impairment	147 (32.7)	108 (34.5)	66 (36.3)		
Moderate cognitive impairment	105 (23.3)	59 (18.8)	36 (19.8)		
Severe cognitive impairment	35 (7.8)	21 (6.7)	13 (7.1)		
RBANS					
Immediate memory	53 (44,65)	53 (44,69)	53 (49,73)	3.731	0.155
Visuospatial skills	84 (66,92)	81 (66,96)	81 (64,92)	3.399	0.183
Language	80.34 (12.81)	81.17 (13.23)	80.47 (15.27)	0.282	0.754
Attention	78.62 (14.76)	78.19 (15.52)	78.31 (15.57)	0.06	0.942
Delayed memory	60 (48,80)	60 (48,83)	68 (48,84)	2.875	0.238
Sum of scales	360.5 (63.66)	363.75 (72.6)	360.56 (62.27)	0.191	0.826
Total score	66 (54,75)	64 (54,78)	69 (55,78)	0.327	0.849

Note: MMSE: Mini-mental State Examination repeated; RBANS: battery for the assessment of neuropsychological status.

**Table 3 brainsci-13-01245-t003:** Risk factors for overweight and obesity.

	B	Standard Error	Wald	*p*	OR	95%CI Lower	95%CI Upper
Age	−0.012	0.006	3.498	0.061	0.988	0.976	1.001
Educational level	−0.122	0.081	2.284	0.131	0.885	0.756	1.037
Gender	0.043	0.178	0.058	0.809	1.044	0.736	1.481
Diabetes	0.637	0.209	9.269	0.002	1.891	1.255	2.849
SumPANSP	0.030	0.015	3.863	0.049	1.030	1.000	1.061
Systolic blood pressure	0.011	0.008	1.781	0.182	1.011	0.995	1.026

Note: educational level: 1—elementary school; 2—junior high school; 3—secondary high school; 4—undergraduate; 5—master’s degree; 6—doctorate; 7—uneducated.

**Table 4 brainsci-13-01245-t004:** Risk factors for higher BMI.

	B	Standard Error	Beta	*t*	*p*	Tolerance	VIF
Gender	0.024	0.035	0.023	0.703	0.482	0.943	1.061
Age	−0.004	0.001	−0.098	−2.830	0.005	0.859	1.164
Educational level	−0.037	0.016	−0.073	−2.261	0.024	0.980	1.020
SumPANSP	0.008	0.003	0.084	2.552	0.011	0.950	1.052
Diabetes	0.133	0.049	0.092	2.721	0.007	0.896	1.117
Hypertension	0.083	0.047	0.060	1.752	0.080	0.886	1.128

Note: VIF: Variance Inflation Factor; educational level: 1—elementary school; 2—junior high school; 3—secondary high school; 4—undergraduate; 5—master’s degree; 6—doctorate; 7—uneducated.

## Data Availability

The data presented in this study are available upon request from the corresponding author.

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
