# Peer review of "Overweight or Obesity among Chinese Han People with Schizophrenia: Demographic, Clinical and Cognitive Correlations"

_brainsci, 2023, doi:10.3390/brainsci13091245_

Round 1

Reviewer 1 Report

The authors performed cross sectional study to find a relation between overweight and obesity, clinical features and the cognitive performance of Chinese patients with chronic schizophrenia who had overweight or obesity. This study has numerous methodological flaws; merely mentioning the limits of the study without attempting to address them is insufficient.

1-      According to the authors, patients with overweight and obesity were younger and had less education. As a result, because it was designed to identify and characterize aberrant cognitive decline in older adults and as a neuropsychological screening battery for younger patients, RBANS is not a viable test for assessing cognitive performance in those patients.

2-      Because of features of linguistic and mathematics assessment, the Mini-mental State Examination is biased against those with low education, and the author indicated that patients with overweight and obesity were younger and had less education.

3-      There is no information available about medication use, dosing, or the effect on body weight. All the data were collected from chronic Chinese patients who had been taking antipsychotics for a long time, and the dose and specifics of antipsychotic treatment might impact the results.

4-      What about the disease duration of the patients involved?

Minor English editing is required.

Author Response

We feel great thanks for your professional review work on our article. As you are concerned, there are several problems that need to be addressed. According to your nice suggestions, we have made some corrections to our previous draft, the detailed corrections are listed below.

The authors performed cross sectional study to find a relation between overweight and obesity, clinical features and the cognitive performance of Chinese patients with chronic schizophrenia who had overweight or obesity. This study has numerous methodological flaws; merely mentioning the limits of the study without attempting to address them is insufficient.

  • According to the authors, patients with overweight and obesity were younger and had less education. As a result, because it was designed to identify and characterize aberrant cognitive decline in older adults and as a neuropsychological screening battery for younger patients, RBANS is not a viable test for assessing cognitive performance in those patients.

Reply: We sincerely appreciate the valuable comments. As you suggested, RBANS was designed to identify and characterize aberrant cognitive decline in older adults. In the present study, average age in normal group, overweight group and obesity group were 48±13.2, 47±11.6, and 45±12.3 respectively. Although our results showed that patients with overweight group and obesity were younger, the mean age of the patients in all three groups was greater than 45 years old. So we thought these patients may can be used with RBANS. However, we agree that this is a potential limitation of the study. We have added this as a limitation in the revised manuscript, showing as: Fifthly, because RBANS was designed to identify and characterize aberrant cognitive decline in older adults, it may be not the best choice for assessing cognitive performance in this study (Since our included patients were predominantly middle-aged).

  • Because of features of linguistic and mathematics assessment, the Mini-mental State Examination is biased against those with low education, and the author indicated that patients with overweight and obesity were younger and had less education. 

Reply: Thanks for your suggestion. We agree that this is a potential limitation of the study. We compared the MMSE scores of overweight, obese and normal schizophrenic patients using education years as adjusting factor. We got the same results that there was no significant difference in MMSE scores between the three groups (P>0.05).Nevertheless, we do agree that this is a possible limitation of the present study and we have added this as a limitation in the revised manuscript, showing as: Also, Because of features of linguistic and mathematics assessment, the MMSE is biased against those with low education.

  • There is no information available about medication use, dosing, or the effect on body weight. All the data were collected from chronic Chinese patients who had been taking antipsychotics for a long time, and the dose and specifics of antipsychotic treatment might impact the results.

Reply: We think this is an excellent suggestion. We have added the information available about medication use in the revised manuscript, which is showing as:

Antipsychotics administered to patients consisted of: clozapine (n = 239), risperidone (n = 194), olanzapine (n = 78), aripiprazole (n = 56), quetiapine (n = 31), sulpiride (n = 20), amisulpride (n = 14), ziprasidone (n = 7), perphenazine (n = 6), chlorpromazine (n = 4), and other (n = 8). The usage of antipsychotics in eight patients was not documented in this study.

Daily doses of antipsychotics were converted to equivalent doses of chlorpromazine. The daily dosage of antipsychotics in all patients was 249.2±124.6 mg/day. The daily dosage of antipsychotics in the normal, overweight and obese groups were 250(145,410), 290(150,404.4) and 240(125,400)mg/day respectively (Z=2.859, P=0.239).

  • What about the disease duration of the patients involved?

Reply: As suggested, we have added the information about the disease duration of the patients involved. The disease duration of the patients in the normal, overweight and obese groups were 22(12,31), 21(12,30), and 19(11,30) respectively. (Z=4.946, P=0.084)

Minor English editing is required.

Reply: Thanks for your suggestion. We have tried our best to polish the language in the revised manuscript.

Reviewer 2 Report

The manuscript provides a nuanced understanding of the prevalence and clinical characteristics of overweight and obesity among individuals with schizophrenia, specifically within the Chinese Han population. This work is important as it could guide more personalized patient care, aiding in developing more effective treatment strategies and potentially leading to better patient outcomes.

These are constructive suggestions and overall the manuscript is well-structured with a clear focus. It addresses an important and under-researched area.

MINOR Grammar and style ISSUES

TITLE

1.     The title is specific, but could be more concise and meaningful. Try "Overweight and Obesity among Chinese Han Schizophrenia Patients: Demographic, Clinical, and Cognitive Correlations".

ABSTRACT

2.     The abstract is mostly well-written, but it lacks information on methods used in the study. Adding a brief description of the methodology could enhance the abstract.

3.     Moreover, since this is a manuscript dealing with a particular population (Chinese Han), it would be useful to mention this fact in the abstract to better contextualize the study

4.     You should mention the statistical methods used in your study in the abstract for clarity.

5.     When mentioning statistical significance (P<0.05), explain which groups you are comparing or variables you're associating.

6.     You could consider presenting key results with more specific statistics.

7.     Reword "Logistics regression analysis" to "Logistic regression analysis".

8.     The phrase "In China, patients with chronic schizophrenia have a high prevalence of overweight and obesity." seems redundant, since this point was already mentioned in the first sentence.

Keywords

9.     (line 27): The chosen keywords should be refined. For example, “discrepancies” doesn’t appear to be a relevant keyword for the current study. Considering the use of terms like "Chinese Han" or "demographics" could be more effective..

 Introduction:

10.  Line 37, "more likely to combined" should be "more likely to combine".

11.  It would be better to present a clear research gap this study seeks to address. before mentioning the objectives of your study.

12.  it would be helpful to explicitly state the research question and study's objectives.

 methods:

1.     Line 75: Change "We included chronic inpatients during December 2006 and December 2008" to "We included chronic inpatients admitted between December 2006 and December 2008".

2.     Line 78: Be clear about who is giving the informed consent (presumably the patients or their legal guardians).

3.     Line 80: "the duration of illness >=5 years" can be more clearly written as "a minimum illness duration of 5 years".

4.     Line 82: Specify which antipsychotics were used.

5.     Line 86: The phrase "calibrated to 0.1 kg using an electronic scale [eb9003l, xiangshan, china]" seems to be misplaced and should be moved to the methods section where you explain how you measured weight and height.

 RESULTS

6.       Line 136-139: The authors state that "those with overweight and obesity were younger, had less education, higher waist and hip, more likely to be female and have higher rate of diabetes. Moreover, patients with overweight or obesity had higher sumPANSP than those without overweight and obesity." This is a complex sentence with multiple findings. It could be broken down into several sentences for clarity and ease of reading.

7.       Lines 134-135: Ensure the provided values in parentheses follow the correct order i.e., (observed/expected) or vice versa.

8.       Line 144: Clarify the specifics of the statistical test used, and provide a measure of effect size. Just reporting p values doesn't give a complete picture of the significance of the results.

9.       In general, be more explicit when reporting results. For example, include mean or median scores for PANSS, and more information on demographics (e.g., education levels)

DISCUSSION

10.   Line 156: The transition between results and discussion ("4. Discussion 156") seems abrupt. A concluding sentence at the end of the results section could better prepare the reader for the discussion.

11.   Lines 157-159: The authors state that this is the "first" study of this kind, which should be verified with a thorough literature review to make sure this is indeed the first study of its kind. Another possibility would be to say this study "adds to the existing literature" rather than it being the "first."

12.   Lines 185-187: The authors write that "we found that Chinese schizophrenia inpatients that had overweight or obesity were more likely to be female. Our study detected higher proportion of women than man, with analyzing the patients with overweight and obesity." The repetition of the same idea can be avoided for clarity and conciseness.

13.   The authors should add  subheadings within the Discussion section to guide the reader through the key points being discussed.

 14.   The authors may also want to consider the use of more active voice and less passive voice for more engaging and direct writing.

 MAJOR ISSUES

15.   It could be useful for this type of study to include a DAG (directed acyclic graph) , to show the relation between variables those can be easily done with the free software daggity. https://www.dagitty.net/  You can see examples of its use in https://doi.org/10.3390/jcm10132859 and https://doi.org/10.3390/jcdd9010025

MATERIAL AND METHODS

16.   Sample Size (line 15): It's important to justify the sample size. Does this sample size have enough power to detect significant differences and effects?

17.   Evaluation Methods (lines 16-18): There should be more information about the validity and reliability of the Mini-Mental State Examination (MMSE), and the Repeated Battery for Evaluation of the Neuropsychological Status (RBANS) in the methods section.

18.   Subjects' Inclusion and Exclusion Criteria (lines 79-84): The criteria are clearly defined but further clarification on how these were assessed would be beneficial. Also, it would be helpful to justify the chosen age range and duration of illness.

19.   Measurement of Variables (lines 96-112): The description of how different variables were measured could be clearer. More detailed explanation of scales and indexes used, especially for non-specialist readers, would be helpful.

20.   Statistical Analysis (lines 115-127): This section could benefit from more detail, such as how missing data was handled, which specific variables were controlled for, and the rationale behind the chosen statistical tests. Moreover, justifying Bonferroni correction for multiple comparisons would be beneficial as it is a conservative method.

 Results

21.   In material and methods the author said that they used lineal regression, but reading the results it is not clear if the used simple o multiple lineal regression.

22.   In lines 151-154, the interpretation of the logistic regression and linear regression results may be misleading. For example, the OR for educational years is reported as 1.05, which is close to 1 and suggests almost no association. But it's declared as a significant factor for overweight and obesity. It might be more useful to discuss why these parameters showed significance despite having little impact, or to reconsider the model used.

23.   The authors says  in lines 151-152 “educational years (OR=1.05, 151 95%CI=1.000-1.103, P=0.049)). “ but this does not agree with the table. Where educational years OR = 0.952  (95% CI 0.907 1). Authors should double check their data and correct either the text or the table.

24.   Authors should include a table containing the or the simple multiple lineal regression models. They should include in the text, the R and the R2. In addition of B, it should add also Beta standardized parameters that appears in the SPSS outpu, because when different units are used like age, or PANSP , or systolic pressure it allows to know the importance of each of them in the model.

DISCUSSION

25.   Lines 156-163: The authors should interpret the results in the context of existing literature, stating whether their findings are consistent with those of other studies.

26.   Line 161-163: Be careful not to overinterpret the data. For example, the differences in education years are statistically significant, but the actual differences are quite small. Discuss whether these differences are clinically meaningful. The small size of the difference could because you measure the increase in the OR for each year of education. 1.05 would be for 1year. for 6 year would be 1.05**6=1.34 , for 10years=1.05**10= 1.62, etc  If you have used other unit eg education in decades , the OR could be higher. 

27.   Lines 164-174: Discuss why the results from the present study are different from those of other studies (e.g., sample characteristics, different measures used).

The English have to be reviewed, the manuscript is an easy read, but there are minor grammatical errors eg: There is a grammatical error in the second sentence of  the Abstract: ("Schizophrenia patients are more likely to combined..."), it should be corrected to: "Schizophrenia patients are more likely to combine...". Overall, this manuscript could also benefit from a thorough proofreading to correct minor grammatical errors and ensure consistency in terminology and presentation of data.

Author Response

We feel great thanks for your professional review work on our article. As you are concerned, there are several problems that need to be addressed. We tried our best to improve the manuscript and made some changes to the manuscript. According to your nice suggestions, we have made extensive corrections to our previous draft, the detailed corrections are listed below. These changes will not influence the content and framework of the paper. And here we did not list the changes but mark in yellow in the revised paper.

The manuscript provides a nuanced understanding of the prevalence and clinical characteristics of overweight and obesity among individuals with schizophrenia, specifically within the Chinese Han population. This work is important as it could guide more personalized patient care, aiding in developing more effective treatment strategies and potentially leading to better patient outcomes.

These are constructive suggestions and overall the manuscript is well-structured with a clear focus. It addresses an important and under-researched area.

MINOR Grammar and style ISSUES

TITLE

  1. The title is specific, but could be more concise and meaningful. Try "Overweight and Obesity among Chinese Han Schizophrenia Patients: Demographic, Clinical, and Cognitive Correlations".

Reply: Thank you for your suggestion. We have revised the title as "Overweight and Obesity among Chinese Han Schizophrenia Patients: Demographic, Clinical, and Cognitive Correlations".

ABSTRACT

  1. The abstract is mostly well-written, but it lacks information on methods used in the study. Adding a brief description of the methodology could enhance the abstract.

Reply: Thank you for your suggestion. We have added a brief description of the methodology in the abstract, which is showing as: All patients were evaluated with the positive and negative syndrome scale, the Mini-Mental State Examination (MMSE) and the repeated battery for evaluation of the neuropsychological status (RBANS) scale. We collected demographic and clinical data using self-reported questionnaires. We divided patients into normal weight (BMI<24 kg/m2), overweight (24≤BMI<28 kg/m2) and obese (≥28 kg/m2) groups according to the working group on obesity in China (WGOC) criteria. We compared the clinical data between the three groups and then conducted binary logistic regression and linear regression to assess variables that were significantly associated with overweight and obesity and higher BMI.

  1. Moreover, since this is a manuscript dealing with a particular population (Chinese Han), it would be useful to mention this fact in the abstract to better contextualize the study

Reply: Thank you for your suggestion. We have mentioned Chinese Han population in the revised manuscript.

  1. You should mention the statistical methods used in your study in the abstract for clarity.

Reply: Thank you for your suggestion. We have mentioned the statistical methods used in the study, showing as : We divided patients into normal weight (BMI<24 kg/m2), overweight (24≤BMI<28 kg/m2) and obese (≥28 kg/m2) groups according to the working group on obesity in China (WGOC) criteria. We compared the clinical data between the three groups and then conducted binary logistic regression and linear regression to assess variables that were significantly associated with overweight and obesity and higher BMI.

  1. When mentioning statistical significance (P<0.05), explain which groups you are comparing or variables you're associating.

Reply: Thank you for your suggestion. We have added the details of the groups we are comparing or variables we're associating in the manuscript, showing as: Of the sample, 324 (32.9%) and 191 (19.4%) patients had overweight and obesity respectively. Patients who had overweight and obesity were younger, had less education, had higher waist and hip circumferences, higher rates of diabetes and higher sumPANSP score (compared with patients in the normal group, P<0.05). There were more female patients with obesity (compared with patients in the normal and overweight group, P<0.05). 

You could consider presenting key results with more specific statistics.

Reply: Thank you for your suggestion. We have modified the abstract with more specific statistics in the results, showing as: Patients who had overweight and obesity were younger, had less education, had higher waist and hip circumferences, higher rates of diabetes and higher sumPANSP score (compared with patients in the normal group, P<0.05). There were more female patients with obesity (compared with patients in the normal and overweight group, P<0.05). Logistic regression analysis indicated that overweight and obesity were associated with sumPANSP(OR=1.03, 95%CI=1-1.061, P=0.049) and diabetes (OR=1.891, 95%CI=1.255-2.849, P=0.002). Further linear regression showed that age (B=-0.004, t=-2.83, P=0.005), educational level (B=-0.037, t=-2.261, P=0.024), diabetes (B=0.133, t=2.721, P=0.007) and sumPANSP (B=0.008, t=2.552, P=0.011) were risk factors for higher BMI.

  1. Reword "Logistics regression analysis" to "Logistic regression analysis".

Reply: We were really sorry for our careless mistakes. Thank you for your reminder. We have changed the reword "Logistics regression analysis" to "Logistic regression analysis".

  1. The phrase "In China, patients with chronic schizophrenia have a high prevalence of overweight and obesity." seems redundant, since this point was already mentioned in the first sentence.

Reply: Thank you for your reminder. We have removed this sentence in the revised manuscript.

Keywords 

  1. (line 27): The chosen keywords should be refined. For example, “discrepancies” doesn’t appear to be a relevant keyword for the current study. Considering the use of terms like "Chinese Han" or "demographics" could be more effective..

Reply: We think this is an excellent suggestion. As suggested, we have changed the keywords as “Schizophrenia; Chinese Han; demographics; overweight; obesity; cognition”.

 Introduction:

  1. Line 37, "more likely to combined" should be "more likely to combine".

Reply: We sincerely thank you for your careful reading. We have modified the sentence "more likely to combined" to be "more likely to combine" in the revised manuscript.

  1. It would be better to present a clear research gap this study seeks to address. before mentioning the objectives of your study.

Reply: Thank you for your suggestion. As suggested, we have presented a research gap this study seeks to address in the revised manuscript in introduction section, which is showing as: However, whether people with schizophrenia with overweight or obesity have serious cognitive decline is still unclear. And we have no idea about the prevalence and risk factors of overweight and obesity in Chinese Han people with schizophrenia. Therefore we performed the repeated battery for evaluation of the neuropsychological status (RBANS) to confirm the hypothesis that worse cognitive performance would be found in people with schizophrenia with overweight or obesity in this study.

  1. it would be helpful to explicitly state the research question and study's objectives.

Reply: Thank you for your suggestion. As suggested, we have added the research question and study's objectives in the introduction section, showing as: The study presented here ought to explore (1) the prevalence of overweight and obesity in Chinese Han people with schizophrenia, (2) cognitive function and the clinical correlates in these specific patients. We proposed the following hypotheses (1) the high prevalence of overweight and obesity in Chinese Han people with schizophrenia; (2) people with schizophrenia with overweight and obesity exhibit greater cognitive impairment cognition than people with schizophrenia with normal BMI. (3) there would be some clinical correlates and risk factors in people with schizophrenia with overweight and obesity.

 methods:

  1. Line 75: Change "We included chronic inpatients during December 2006 and December 2008" to "We included chronic inpatients admitted between December 2006 and December 2008".

Reply: We sincerely thank you for your careful reading. As suggested, we have changed this in the revised manuscript.

 Line 78: Be clear about who is giving the informed consent (presumably the patients or their legal guardians).

Reply: Thank you for your suggestion. The informed consent was got from the patients, as we included patients who were competent to consent based on an evaluation of their comprehension of the consent form.

  1. Line 80: "the duration of illness >=5 years" can be more clearly written as "a minimum illness duration of 5 years".

Reply: Thank you for your suggestion. We have changed this in the revised manuscript.

  1. Line 82: Specify which antipsychotics were used.

Reply: We think this is an excellent suggestion. We have added the information available about medication use in the revised manuscript, which is showing as:

Antipsychotics administered to patients consisted of: clozapine (n = 239), risperidone (n = 194), olanzapine (n = 78), aripiprazole (n = 56), quetiapine (n = 31), sulpiride (n = 20), amisulpride (n = 14), ziprasidone (n = 7), perphenazine (n = 6), chlorpromazine (n = 4), and other (n = 8). The usage of antipsychotics in eight patients was not documented in this study.

Daily doses of antipsychotics were converted to equivalent doses of chlorpromazine. The daily dosage of antipsychotics in all patients was 249.2±124.6 mg/day. The daily dosage of antipsychotics in the normal, overweight and obese groups were 250(145,410), 290(150,404.4) and 240(125,400) mg/day respectively (Z=2.859, P=0.239).

  1. Line 86: The phrase "calibrated to 0.1 kg using an electronic scale [eb9003l, xiangshan, china]" seems to be misplaced and should be moved to the methods section where you explain how you measured weight and height.

Reply: Thank you for your suggestion. We have changed this sentence to the 2.2. Section of methods where we explain how we measured weight and height.

 RESULTS

  1. Line 136-139: The authors state that "those with overweight and obesity were younger, had less education, higher waist and hip, more likely to be female and have higher rate of diabetes. Moreover, patients with overweight or obesity had higher sumPANSP than those without overweight and obesity." This is a complex sentence with multiple findings. It could be broken down into several sentences for clarity and ease of reading.

Reply: It is really true as Reviewers suggested that this is a complex sentence with multiple findings for not ease of reading. We have broken down into several sentences in the revised manuscript showing as: Compared with patients without overweight or obesity, people with schizophrenia with overweight and obesity were younger, had lower education and had higher waist and hip (both P <0.01; Bonferroni corrected both P< 0.05). And people with schizophrenia with overweight and obesity were more likely to be female (χ2=25.191, P<0.001) and have a higher rate of diabetes (χ2=6.311, P=0.043) (only sex survived after Bonferroni corrected). Moreover, people with schizophrenia with overweight or obesity had higher sumPANSP than those without overweight and obesity (obesity: 17.74±5.49 vs overweight: 16.24±5.8 vs normal: 15.71±4.87; d=0.203, 95%CI=0.078-0.329; P <0.001; Bonferroni corrected both P< 0.05). 

Lines 134-135: Ensure the provided values in parentheses follow the correct order i.e., (observed/expected) or vice versa.

Thanks you for your kind reminder. We have confirmed that the provided values in parentheses follow the correct order.

  1. Line 144: Clarify the specifics of the statistical test used, and provide a measure of effect size. Just reporting p values doesn't give a complete picture of the significance of the results.

Reply: Thanks you for your kind reminder. We have added more specifics of the statistics in the revised manuscript. (RBANS: Non-parametric test and ANOVA test found no significant difference among the three groups between the five RBANS subtests and total scores (P<0.05, adjusting for age, sex and educational years P<0.05).MMSE: Non-parametric test found no significant difference among the three groups (Z=0.431, P=0.806)).

  1. In general, be more explicit when reporting results. For example, include mean or median scores for PANSS, and more information on demographics (e.g., education levels)

Reply: We have added the suggested content to the manuscript of the more explicit results, showing as: The average age of the included patients was 47.2±12.5 years old. The mean educational years and BMI was 9.2±3.2 and 24.8±4.6. The mean values of the total RBANS and MMSE scores for all patients were 78.6±17.4 and 23.2±6.2 respectively.

DISCUSSION

  1. Line 156: The transition between results and discussion ("4. Discussion 156") seems abrupt. A concluding sentence at the end of the results section could better prepare the reader for the discussion.

Reply: We think this is an excellent suggestion. We have added a concluding sentence at the end of the results section, showing as: In this study, the main findings were as follows: (1) 324 (32.9%) and 191 (19.4%) patients had overweight and obesity respectively. Compared with patients without overweight or obesity, people with schizophrenia with overweight and obesity were younger, had lower education and higher waist and hip, more likely to be female and had a higher rate of diabetes. Moreover, people with schizophrenia with overweight or obesity had higher sumPANSP than those without overweight and obesity. (2) No significant differences were found between the five RBANS subtests, total scores and MMSE scores. (3) SumPANSP and diabetes were associated with overweight and obesity in people with schizophrenia. Age, educational level, sumPANSP and diabetes were risk factors for higher BMI.

  1. Lines 157-159: The authors state that this is the "first" study of this kind, which should be verified with a thorough literature review to make sure this is indeed the first study of its kind. Another possibility would be to say this study "adds to the existing literature" rather than it being the "first."

Reply: We agree with your assessment. Accordingly, we have changed "first" to "adds to the existing literature".

  1. Lines 185-187: The authors write that "we found that Chinese schizophrenia inpatients that had overweight or obesity were more likely to be female. Our study detected higher proportion of women than man, with analyzing the patients with overweight and obesity." The repetition of the same idea can be avoided for clarity and conciseness.

Reply: Thanks you for your kind reminder. We have removed the duplicative language in the revised manuscript.

  1. The authors should add  subheadings within the Discussion section to guide the reader through the key points being discussed.

Reply: We agree with your assessment. We have added subheadings within the Discussion section in the revised manuscript.( The prevalence of overweight and obesity; Gender differences in overweight and obesity in people with schizophrenia; PANSS score in people with schizophrenia with overweight and obesity; Cognitive performance in people with schizophrenia with overweight and obesity; Limitations; Conclusions)

  1. The authors may also want to consider the use of more active voice and less passive voice for more engaging and direct writing.

Reply: We sincerely appreciate the valuable comments. We have tried our best to make some changes in the revised paper based on your comments.

 MAJOR ISSUES

  1. It could be useful for this type of study to include a DAG (directed acyclic graph) , to show the relation between variables those can be easily done with the free software daggity. https://www.dagitty.net/  You can see examples of its use in https://doi.org/10.3390/jcm10132859 and https://doi.org/10.3390/jcdd9010025

Reply: We think this is an excellent suggestion. We have added a DAG in the statistics section, showing as: DAG (directed acyclic graph) is useful to show the relation between variables in the cross-sectional study. The DAG makes it possible to draw theoretical diagrams of the links between variables and to identify which variables should be controlled in a multivariate model, thus preventing bias. We used DAG (Figure 1) to explore the association of schizophrenia and overweight and obesity and potential confounding factors. Using this graph, the software Dagitty indicated that the possible cofounders that should be adjusted were age and sex.

MATERIAL AND METHODS

  1. Sample Size (line 15): It's important to justify the sample size. Does this sample size have enough power to detect significant differences and effects?

Reply: Thanks for the mention. We calculated the sample size. This was a cross-sectional study and we assumed that the prevalence of overweight or obesity in patients with schizophrenia was 25% based on previous literature data. We obtain a sample size of 832 with a two-sided a as 0.05 and a tolerance error of 3% . Finally a minimum of 924 patients with schizophrenia was required when taking into account a 10% loss to follow-up rate. Our study included 985 SCZ patients, so this sample size has enough power to detect significant differences and effects.

  1. Evaluation Methods (lines 16-18): There should be more information about the validity and reliability of the Mini-Mental State Examination (MMSE), and the Repeated Battery for Evaluation of the Neuropsychological Status (RBANS) in the methods section.

Reply: We think this is an excellent suggestion. We have added more information about the validity and reliability of the Mini-Mental State Examination (MMSE), and the Repeated Battery for Evaluation of the Neuropsychological Status (RBANS) in the methods section, showing as: Mini-mental state examination (MMSE) was used for cognition assessment. MMSE includes 5 subsections (orientation, regitration and recall, attention can calculation, language, and praxis). MMSE scores vary between 0 - 30, with lower scores suggesting poorer cognitive ability. Every question carries three possible answers: correct, incorrect and unanswerable. We calculate the unanswerable as the incorrect answer. MMSE uses a cutoff of 25 as a diagnosis of mild cognitive impairment (MCI)(Lim and Loo, 2018).

Two psychologists conducted the Repeatable Battery for the Assessment of Neuropsychological Status (RBANS, Form A) to assess cognition. Except for effective screening for dementia, it can also be conducted for a variety of disorders, including schizophrenia(Li et al., 2020). RBANS includes 12 subtests which resulted in five age-adjusted index scores and a total score. The five sections consist of attention, language, visuospatial/constructional, immediate memory, and delay memory. RBANS is utilized for assessment when the patient is stable and does not present with psychiatric symptoms(Hui et al., 2016). Zhong et al. showed its good clinical utility and dependability in schizophrenic patients(Zhong et al., 2013). RBANS showed excellent clinical efficacy and test-retest reliability in both people with schizophrenia and the general population.

The scales were evaluated by trained psychiatrists. The inter-rater correlation coefficient of the scales between the evaluator were all greater than 0.8. They assessed patients' cognitive performance with RBANS on the day of or the following day of blood draws.

  1. Subjects' Inclusion and Exclusion Criteria (lines 79-84): The criteria are clearly defined but further clarification on how these were assessed would be beneficial. Also, it would be helpful to justify the chosen age range and duration of illness.

Reply: Thanks for the mention. We have further elucidated on how these were assessed in the revised manuscript. We chose the age criterion of 18-70 to exclude children or extremely old patients. We included patients with a minimum illness duration of 1 years was to exclude patients with a first episode of schizophrenia in the recent years. These were designed to minimize the interference of other factors.

  1. Measurement of Variables (lines 96-112): The description of how different variables were measured could be clearer. More detailed explanation of scales and indexes used, especially for non-specialist readers, would be helpful.

Reply: We have added the suggested content to the manuscript to explain of scales and indexes used in the study.

  1. Statistical Analysis (lines 115-127): This section could benefit from more detail, such as how missing data was handled, which specific variables were controlled for, and the rationale behind the chosen statistical tests. Moreover, justifying Bonferroni correction for multiple comparisons would be beneficial as it is a conservative method.

Reply: We have added the suggested content to the manuscript, showing as: We performed Shapiro-Wilk test and Q-Q plots to test the normality, and conducted Levene test to confirm the equality of variances. Patients were divided into the normal group, overweight group and obesity group according to BMI. We expressed the incidence of overweight and obesity among people with schizophrenia as a percentage by the chi-square (χ2) test to compare the incidence between male and female patients. Analysis of variance and χ2 were performed to compare the variations in demographic and clinical characteristics between the three groups. Missing values are interpolated using mean values. The Bonferroni correction was performed to adjust for multiple tests. Multivariate analysis of covariance was conducted to investigate differences in cognitive status on the five and the total index scores of the RBANS, while adjusting for the potential confounding parameters(age, sex and educational years). Binary logistic regression was performed to assess variables that were significantly associated with overweight and obesity. Odds ratios (OR) resulted from logistic regression analyses to compare overweight and obesity among people with schizophrenia after correcting for associated variables. To better clarify the link between BMI levels and statistically significant indicators, we conducted multiple linear regression to assess variables that were significantly correlated with higher BMI. All analysis was performed in SPSS version 25.0. The statistical significance was set with P<0.05.

 Results

  1. In material and methods the author said that they used lineal regression, but reading the results it is not clear if the used simple o multiple lineal regression.

Reply: We are very sorry for our negligence of that. We used multiple lineal regression in this study.

  1. In lines 151-154, the interpretation of the logistic regression and linear regression results may be misleading. For example, the OR for educational years is reported as 1.05, which is close to 1 and suggests almost no association. But it's declared as a significant factor for overweight and obesity. It might be more useful to discuss why these parameters showed significance despite having little impact, or to reconsider the model used.

Reply: We sincerely appreciate the valuable comments. As recommend, we re-analyze the data based on educational level (Elementary school, middle school, high school...), the regression results found that education level was not a risk factor for overweight and obesity in schizophrenics (OR=0.885, 95%CI=0.756-1.037, P=0.131). Only diabetes and sumPANSP were risk factors.

discussion

  1. The authors says  in lines 151-152 “educational years (OR=1.05, 151 95%CI=1.000-1.103, P=0.049)). “ but this does not agree with the table. Where educational years OR = 0.952  (95% CI 0.907 1). Authors should double check their data and correct either the text or the table.

Reply: We feel sorry for our carelessness. We have modified this mistake and double checked our data and correct either the text or the table.

  1. Authors should include a table containing the or the simple multiple lineal regression models. They should include in the text, the R and the R2. In addition of B, it should add also Beta standardized parameters that appears in the SPSS outpu, because when different units are used like age, or PANSP , or systolic pressure it allows to know the importance of each of them in the model.

Reply: Thank you for your suggestion. We have included a table containing the multiple lineal regression models in the revised manuscript, showing as:

Table 4 risk factors for higher BMI

B

Standard error

Beta

t

P

Tolerance

VIF

Gender

0.024

0.035

0.023

0.703

0.482

0.943

1.061

Age

-0.004

0.001

-0.098

-2.830

0.005

0.859

1.164

Educational level

-0.037

0.016

-0.073

-2.261

0.024

0.980

1.020

SumPANSP

0.008

0.003

0.084

2.552

0.011

0.950

1.052

Diabetes

0.133

0.049

0.092

2.721

0.007

0.896

1.117

Hypertension

0.083

0.047

0.060

1.752

0.080

0.886

1.128

DISCUSSION

  1. Lines 156-163: The authors should interpret the results in the context of existing literature, stating whether their findings are consistent with those of other studies.

Reply: Thank you for your suggestion. We summarize the main findings in the discussion and then we interpret the results separately. Considering the Reviewers’ suggestion, we have added the content of whether our findings are consistent with other studies in the discussion.

  1. Line 161-163: Be careful not to overinterpret the data. For example, the differences in education years are statistically significant, but the actual differences are quite small. Discuss whether these differences are clinically meaningful. The small size of the difference could because you measure the increase in the OR for each year of education. 1.05 would be for 1year. for 6 year would be 1.05**6=1.34 , for 10years=1.05**10= 1.62, etc  If you have used other unit eg education in decades , the OR could be higher. 

Reply: We agree with your assessment. Your suggestion really means a lot to us. We did overinterpret the data undeniably. Yes, it would be more understandable if we classified the educational years as educational level. We re-analyze the data based on educational level (Elementary school, middle school, high school...), the regression results found that education level was not a risk factor for overweight and obesity in schizophrenics. (OR=0.885, 95%CI=0.756-1.037, P=0.131).

  1. Lines 164-174: Discuss why the results from the present study are different from those of other studies (e.g., sample characteristics, different measures used).

Reply: We sincerely appreciate the valuable comments. We have added this point to the discussion section, showing as: The reasons that our results are different from other studies may be sample characteristics and different measures used. For example, our study population focused on people with schizophrenia with a mean age of 47 years, implying an included population that is predominantly middle-aged. At the same time, age is an important factor in overweight and obesity(Wang et al., 2020). Being obese or overweight may increase disease burden, increase stigma, decrease self-esteem and social functioning, and decrease self-management behaviors like adhering to medication regimens(Fleischhaker et al., 2008). The discrepancies in overweight and obesity rates might also be due to other reasons such as genetics, culture, environment and medication treatment (Qasim et al., 2018; Tessier et al., 2019).

Reviewer 3 Report

The present study investigates the socio-demographic, clinical and cognitive correlates of overweight and obesity in people living with schizophrenia in a sample of 908 inpatients recruited in China.

The study is of scientific interest, focusing on topics of considerable clinical importance, and features a large sample size. The statistical analyses are overall appropriate for the study design.

However, several significant issues have to be addressed in order to reach a quality level that is adequate for publication.

General Issues:

-The term “schizophrenia patient” should be replaced with the more adequate “person (or people, or individual) with (or diagnosed with, or living with) schizophrenia” throughout the whole manuscript.

-The use of English language is quite awkward, and some sentences are not clear and difficult to read. The manuscript should be submitted to professional proofreading or, at least, should be checked by a native English speaker to improve clarity.

Introduction:

-Several sentences in the introduction section do not have recent references supporting them. Overall, alongside the current references, more recent literature should be cited in the introduction section.

-The study aims and primary hypotheses should be explained more clearly at end of the introduction section.

Methods:

-The methods section reports that data were obtained in December 2006 and December 2008. If this is not a typo, the analyzed data are now more that 16 years old: this is a considerable cause for concern. In fact, it should be explained in detail how data was handled and kept in this long period of time while ensuring the security of participants’ privacy. Furthermore, it should be discussed why this dataset was analyzed after 16 years for the present investigation.

-Collinearity diagnostics for the multiple regression analyses (Variance Inflation Factor and tolerance) should be performed and included in the manuscript results.

Results:

-Variables and comparisons that emerged as significant in the analyses should be highlighted in the tables (for instance by using a bold font to highlight significant p-values, or by adding asterisks to significant p-values).

Discussion:

-As cognitive outcomes represent one of the topics on which the manuscript is focused, a paragraph detailing available and effective treatments for cognitive impairment in people living with schizophrenia should be included in the discussion section (see Vita A et al., European Psychiatric Association guidance on treatment of cognitive impairment in schizophrenia. Eur Psychiatry. 2022. 65(1):e57. doi: 10.1192/j.eurpsy.2022.2315).

In particular, the combination of cognitive remediation and physical exercise should be mentioned in this context, as it could provide valuable improvements in both cognitive and metabolic outcomes such as overweight and obesity (see Deste G et al., Impact of Physical Exercise Alone or in Combination with Cognitive Remediation on Cognitive Functions in People with Schizophrenia: A Qualitative Critical Review. Brain Sci. 2023. 13(2):320. doi: 10.3390/brainsci13020320).

-The large size of the included sample and the use of validate measures to assess clinical and cognitive characteristics of participants could be mentioned as points of strength of the study.

-The use of English language is quite awkward, and some sentences are not clear and difficult to read. The manuscript should be submitted to professional proofreading or, at least, should be checked by a native English speaker to improve clarity.

Author Response

We feel great thanks for your professional review work on our article. As you are concerned, there are several problems that need to be addressed. According to your nice suggestions, we have made extensive corrections to our previous draft, the detailed corrections are listed below.

The present study investigates the socio-demographic, clinical and cognitive correlates of overweight and obesity in people living with schizophrenia in a sample of 908 inpatients recruited in China.

The study is of scientific interest, focusing on topics of considerable clinical importance, and features a large sample size. The statistical analyses are overall appropriate for the study design.

However, several significant issues have to be addressed in order to reach a quality level that is adequate for publication.

General Issues:

-The term “schizophrenia patient” should be replaced with the more adequate “person (or people, or individual) with (or diagnosed with, or living with) schizophrenia” throughout the whole manuscript.

Reply: Thank you for your suggestion, we have replaced the term “schizophrenia patient” as “people with schizophrenia” throughout the whole manuscript.

-The use of English language is quite awkward, and some sentences are not clear and difficult to read. The manuscript should be submitted to professional proofreading or, at least, should be checked by a native English speaker to improve clarity.

Reply: Thanks for your suggestion, we feel sorry for our poor writings. We try our best to improve the manuscript and made some changes to the manuscript. We appreciate for your warm work earnestly and hope that the correction will meet with approval.

Introduction:

-Several sentences in the introduction section do not have recent references supporting them. Overall, alongside the current references, more recent literature should be cited in the introduction section.

Reply: As suggested, we have added more references to support our idea in the introduction section.

-The study aims and primary hypotheses should be explained more clearly at end of the introduction section.

Reply: We think this is an excellent suggestion. We have added the study aims and primary hypotheses in the introduction, showing as: The study presented here ought to explore (1) the prevalence of overweight and obesity in Chinese Han people with schizophrenia, (2) cognitive function and the clinical correlates in these specific patients. We proposed the following hypotheses (1) the high prevalence of overweight and obesity in Chinese Han people with schizophrenia; (2) people with schizophrenia with overweight and obesity exhibit greater cognitive impairment cognition than people with schizophrenia with normal BMI. (3) there would be some clinical correlates and risk factors in people with schizophrenia with overweight and obesity.

Methods:

-The methods section reports that data were obtained in December 2006 and December 2008. If this is not a typo, the analyzed data are now more than 16 years old: this is a considerable cause for concern. In fact, it should be explained in detail how data was handled and kept in this long period of time while ensuring the security of participants’ privacy. Furthermore, it should be discussed why this dataset was analyzed after 16 years for the present investigation.

Reply: We feel sorry for our carelessness. Research regarding SCZ patients included in December 2006 and December 2008 was another study conducted in Beijing HuiLong-Guan Hospital and Rong-Jun Hospital in Baoding, Hebei Province. In recent years we have been summarizing data from a number of studies on schizophrenia in which our group has been involved, thus we confusing the timing of the two topics. We apologize for this. The present study included people with SCZ from January to June 2019, and we included in-patients  with chronic schizophrenia in the Guangzhou Huiai Hospital and Wuhan Xinzhou Mental Health Center, which are public psychiatric hospitals in China. 

-Collinearity diagnostics for the multiple regression analyses (Variance Inflation Factor and tolerance) should be performed and included in the manuscript results.

Reply: Thank you for your suggestion. We have included a table containing the multiple lineal regression models in the revised manuscript.

Table 4 risk fators for higher BMI

B

Standard error

Beta

t

P

Tolerance

VIF

Gender

0.024

0.035

0.023

0.703

0.482

0.943

1.061

Age

-0.004

0.001

-0.098

-2.830

0.005

0.859

1.164

Educational level

-0.037

0.016

-0.073

-2.261

0.024

0.980

1.020

SumPANSP

0.008

0.003

0.084

2.552

0.011

0.950

1.052

Diabetes

0.133

0.049

0.092

2.721

0.007

0.896

1.117

Hypertension

0.083

0.047

0.060

1.752

0.080

0.886

1.128

Results:

-Variables and comparisons that emerged as significant in the analyses should be highlighted in the tables (for instance by using a bold font to highlight significant p-values, or by adding asterisks to significant p-values).

Reply: Thank you for your suggestion. We have highlighted variables and comparisons that emerged as significant in the analyses in the tables.

Discussion:

-As cognitive outcomes represent one of the topics on which the manuscript is focused, a paragraph detailing available and effective treatments for cognitive impairment in people living with schizophrenia should be included in the discussion section (see Vita A et al., European Psychiatric Association guidance on treatment of cognitive impairment in schizophrenia. Eur Psychiatry. 2022. 65(1):e57. doi: 10.1192/j.eurpsy.2022.2315).

In particular, the combination of cognitive remediation and physical exercise should be mentioned in this context, as it could provide valuable improvements in both cognitive and metabolic outcomes such as overweight and obesity (see Deste G et al., Impact of Physical Exercise Alone or in Combination with Cognitive Remediation on Cognitive Functions in People with Schizophrenia: A Qualitative Critical Review. Brain Sci. 2023. 13(2):320. doi: 10.3390/brainsci13020320).

Reply: We sincerely appreciate the valuable comments. We have a paragraph detailing available and effective treatments for cognitive impairment in people living with schizophrenia should be included in the discussion section, showing as:

Cognitive performance in people with schizophrenia with overweight and obesity

Cognitive impairment is one of the main symptoms of schizophrenia and is associated with worse outcomes. The burden of anticholinergic and benzodiazepine should be maintained to a minimum taking into account the negative influence on cognitive functioning. In psychosocial interventions, cognitive restoration and physical exercise are suggested to deal with cognitive deficits in schizophrenia. Non-invasive brain stimulation techniques can be considered as an additional therapy(Vita et al., 2022). Both cognitive training (CT) and aerobic exercise show potential for ameliorating cognitive deficits in schizophrenia. The combination of cognitive remediation and physical exercise is recommended as it could provide valuable improvements in cognitive and metabolic outcomes such as overweight and obesity. Also, aerobic exercise performed close to the time of cognitive repair may bring the brain to a state of neuroplasticity readiness, thus promoting cognitive function(Campos et al., 2017). Combining physical exercise with cognitive remediation offered privileged advantages and quicker improvement when compared to cognitive remediation alone(Deste et al., 2023). In the present study, no cognitive performance differences were found between the three subgroups, which were contrary to our assumptions. Previous studies have shown a strong correlation between overweight or obesity and cognition(Leigh and Morris, 2020) (Tanaka et al., 2020), which could result in a difference in the diagnosis of cognition and the inclusion of the population. Therefore, subsequent surveys must be fully addressed.

-The large size of the included sample and the use of validate measures to assess clinical and cognitive characteristics of participants could be mentioned as points of strength of the study.

Reply: We think this is an excellent suggestion. We have mentioned this as points of strength of the study in the discussion revised manuscript, showing as: The strength of the study includes the large size of the included sample and the use of validated measures to assess the clinical and cognitive characteristics of participants.

 -The use of English language is quite awkward, and some sentences are not clear and difficult to read. The manuscript should be submitted to professional proofreading or, at least, should be checked by a native English speaker to improve clarity.

Reply: Thanks for your suggestion, we feel sorry for our poor writings, however, we do invite a friend of us who is a native English speaker to help our article. And we hope the revised manuscript could be acceptable for you.

Round 2

Reviewer 1 Report

The authors respond to my concerns.

Reviewer 3 Report

The Authors responded in a satisfactory manner to all queries and modified the manuscript accordingly.